# *"I feel I have been taken seriously"* Women's experience of greater trochanteric pain syndrome treatment—A nested qualitative study

Jane Andreasen[1,2]*, Angela Fearon[3], Dylan Morissey[4,5], Laura H. Hjørnholm[1], Jens Kristinsson[6], Jens Erik Jorgensen[7], Carsten M. Mølgaard[1,4,8]

1 Department of Physiotherapy and Occupational Therapy, Aalborg University Hospital, Aalborg, Denmark, 2 The Faculty of Medicine, Department of Health Science and Technology, Public Health and Epidemiology Group, Aalborg University, Aalborg, Denmark, 3 UCRISE, Faculty of Health, University of Canberra Hospital, University of Canberra, Bruce ACT, Australia, 4 Sport and Exercise Medicine, Queen Mary University of London, London, United Kingdom, 5 Physiotherapy Department, Barts Health NHS Trust, London, United Kingdom, 6 Department of Orthopaedic Surgery, Aalborg University Hospital, Aalborg, Denmark, 7 Sofiendal Aalborg Health Team, Aalborg SV, Denmark, 8 Department of Clinical Medicine, Aalborg University, Aalborg, Denmark

* jaan@rn.dk

**Data Availability Statement:** Due to the Danish interpretations of the European GDPR legislation, it is unfortunately not possible to make the interview

## Abstract

### Background

Women experiencing greater trochanteric pain syndrome (GTPS) report high levels of pain and reduced quality of life. Exploring how they manage GTPS in a daily life context can provide important knowledge about individual coping strategies. Education, extracorporeal shockwave therapy (ESWT) and exercise have good group level evidence for efficacy in clinical trials and are increasingly used in routine care for patients with GTPS. Exploring women's experiences of such treatment may help understand the mechanisms underpinning these positive results and inform treatment strategies. We therefore aimed to explore how women with GTPS experience and manage their daily life, and their experience of the combined treatment of education, ESWT and exercises.

### Methods

This qualitative study was nested within a cohort study based in a hospital outpatient clinic and a physiotherapy clinic in Denmark assessing the combined treatment of education, ESWT and exercises. Data was collected from eleven women using in-person, individual, semi-structured interviews which were audio recorded. Transcripts were coded and analysed using an inductive thematic analysis approach.

### Findings

Five themes were identified: (1) *Daily life was controlled and structured by pain;* (2) *The condition was acknowledged and taken seriously by treating professionals;* (3) *The participants´ experiences of the intervention–information is key;* (4) *Improved capability and autonomy in*

data public available. However the data are available from Head of Department of Physiotherapy and Occupational Therapy, Aalborg University Hospital, H.B. Pedersen (mail hpe@rn.dk) on reasonable request.

**Funding:** The author(s) received no specific funding for this work.

**Competing interests:** The authors have declared that no competing interests exist.

pain management and (5) *The women´s perspectives on improving and expanding the intervention*. Learning how to manage pain was experienced as the most important element of the program to the women to be able to minimize pain and manage daily life.

## Conclusion

Exploration of how women with greater trochanteric pain syndrome experienced and managed daily hip pain, and how they experienced and adapted to treatment are important novel findings that will inform clinical practice. This new knowledge may be used to inform an individualized patient education, treatment and evaluation strategy for women with the painful and debilitating condition of GTPS.

## Introduction

Greater trochanteric pain syndrome (GTPS) includes disorders of the lateral peri-trochanteric hip area such as trochanteric bursitis, tendinopathy or tears of the gluteus medius and minimus [1–3]. Participants experience pain and tenderness in the region of the greater trochanter, buttock or lateral thigh [4–6]. Greater trochanteric pain syndrome is a common condition with a prevalence of 1.6–3.3 per 1000 in primary care [7, 8] with 36% continuing to report symptoms after 1 year and 29% after five years [7]. In the general population the incidence rate is estimated between 10% and 25% [4, 7, 9–11]. It is more common in women than men, with a higher frequency in post-menopausal women [7].

Participants with GTPS report poor function and high levels of pain and disability [12, 13], lower levels of fulltime employment and lower quality of life than age matched groups [13]. Further, higher pain catastrophizing and depression scores, and lower pain self-efficacy have been demonstrated in participants with severe GTPS [14]. A qualitative study found that participants often had persistent long-term pain and were frustrated that the diagnosis causing their pain was both delayed and confusing [15]. Previous assumptions about GTPS being a mild, self-limiting condition [16, 17] may therefore not be the case and the marked absence of research on interventions and experience is being addressed, as shown by a recent acceleration in the number of randomised controlled trials and qualitative studies [10, 15, 18–22].

There is no clear consensus on best practice for GTPS. High quality studies have shown that combining education and exercise is partially effective for most participants in the short, medium and long term [10] while corticosteroid injections can be beneficial in the short term [9]. Recently, high quality trials have shown extracorporeal shockwave therapy (ESWT) has evidence of efficacy [23–25]. A qualitative study including eleven participants (eight males and three females) found that participants with tendinopathies were largely unaware of the mechanism of action in ESWT and that education on how to handle their condition was seen as important or even more important than ESWT therapy [20]. Further, a scoping review reported that enhanced therapeutic alliance between clinicians and patients with musculoskeletal conditions having physiotherapy may have beneficial effects on treatment fidelity [26]. This is important to successfully addressing GTPS, given that all successful interventions have included elements of education and exercise that require active participation, learning and behavior change. Therefore, aspects of therapeutic alliance as a contributor to the experiences of effect and confidence in managing GTPS are important to explore further.

Recent work carried out to inform trial design has begun to explore participants' understanding of GTPS, their pain and their beliefs regarding activity and exercise [15] but we need

to better understand: how they experience their daily life with GTPS, how participation in a multi-modal treatment program is experienced by the women, and how it may improve their situation regarding pain and functional level. Even though exercise, education and ESWT have good evidence of efficacy in clinical trials and are increasingly used approaches in routine care, the experiential mechanisms explaining treatment outcomes have not been explored. The aim of this study was to explore how women with greater trochanteric pain syndrome, participating in a multi-modal treatment program, experience and manage both their daily life and the combined treatment of education, ESWT and exercise.

## Methods

### Study design

This report was guided by the Consolidated Criteria for Reporting Qualitative Research [27]. This qualitative study was nested within a cohort study (N = 60) conducted in an outpatient hospital clinic and a physiotherapy clinic in Denmark. The cohort study was designed to evaluate treatment outcomes and to identify clinical variables associated with treatment efficacy over six months. The participants were recruited from, and examined at, the hospital outpatient clinic by a physiotherapist specialized in musculo-skeletal physiotherapy (CM) at baseline, and after 12 and 26 weeks. CM provided education about the condition, and advice at each assessment point. The ESWT, exercise therapy and additional education and advice was then performed at a physiotherapy clinic with a second physiotherapist specialized in musculo-skeletal physiotherapy (JEJ).

The "treatment package" included i) a pamphlet and conversation about management of pain and everyday stressors with regards to GTPS with concurrent checks of their understanding during each session, ii) ESWT (once a week for three weeks) followed by six weeks with no intervention and finally iii) six weeks of non-supervised exercise after a thorough introduction. This introduction consisted of an explanation regarding the clinical findings and diagnosis of each participant at inclusion by CM. During the first session by JEJ, which included the ESWT intervention, participants were once more informed, including the provision of written information, about the known mechanisms of how GTPS and associated pain may be managed in everyday life. Participants were educated on how to avoid movements and situations causing pain [10, 28]. The education was repeated and individualised for each patient´s experience, during the following two treatment sessions. At the six-week follow up, participant experience was addressed, and any positive or negative pain experiences were used to facilitate further pain management strategies specific for the individual participant. Thereby moving from a "general" education towards a customized "personal" education specific to the individual participant´s daily context and way of life. Five exercises targeting the lateral muscles of the hip, the stabilizing muscles in the hip and lower back region were introduced (See S1 Appendix). We collected the data for the qualitative study using in-person, individual semi-structured interviews [29].

### Participants

The last 20 participants enrolled in the cohort study were invited to participate at their six month review appointment. Those who consented were invited to interview in accordance with the purposive sampling strategy, designed to ensure participants of different demographic characteristics, age, severity and chronicity of GTPS were represented in order to elicit nuances and sufficient richness of data [29]. The inclusion criteria were women aged between 35 and 70 years, lateral hip pain for at least 3 months, of an intensity of ≥3/10 on an 11-point

**Table 1. Topics included in the interview guide.**

| Interview topics |
| --- |
| Everyday life with GTPS; the experiences related to work life, private life and leisure time before entering the cohort study |
| Experiences and reflections upon their participation in the different elements of the cohort study:<br>• At the baseline assessment and at the follow-up assessments in the cohort study<br>• Advice and guidance<br>• SWT: experience of the treatment, symptoms and pain in relation to treatment<br>• Home exercises |
| Adherence, facilitators, barriers to the elements in the intervention |
| Treatment suggestions for a future treatment plan for GTPS patients; which elements are experienced as important; anything missing in the intervention (in any domain) |
| Their overall experience after participation in the study |

numeric rating scale on most days. Exclusion criteria were severe hip arthritis assessed as a Kellgren-Lawrence score >3 [30].

A female qualitative researcher (JA), not previously involved with the participants, conducted all interviews. JA is an associate professor in public health and experienced with qualitative research and specifically the thematic approach to data analysis. The topic guide for the semi-structured interviews (Table 1) was developed a priori and based on both research literature and the research team's clinical experience [10, 13, 14, 20, 31, 32]. The interview guide included questions and prompts, the latter being used when JA judged it was useful to guide or support the participants. The interviews were performed at the hospital or in the home of the participants depending on the participant's preference. During the Covid-19 pandemic restrictions, one interview was performed using MS Teams teleconferencing platform due to a positive COVID-19 test in the interviewee's family. The interviews lasted between 40 and 70 minutes. All interviews were audiotaped and transcribed verbatim using a predefined transcription guide [29], which was finalized before the analysis was initiated.

## Data analysis

A six-step data-driven thematic analysis was used to analyze the data, as recommended by Braun and Clarke [33, 34]. The theoretical approach was social constructivism to explore insights regarding how people interact with their social world, and focused on the participants' experiences in their particular contexts [33, 35]. This focus can therefore help us to explore how the women themselves construct their knowledge, understanding and management of their GTPS condition in their everyday life. Primarily, an inductive analysis was used [33, 34]. Firstly, initial readings of the interview transcriptions were performed to familiarize with the data. Thereafter, initial codes were generated, and themes identified and reviewed. Themes were then defined and named, and finally, the results were reported. LHH and JA performed the transcriptions, and the subsequent thematic coding was carried out by hand in Microsoft Word. LHH and JA each independently coded the data, and subsequently conducted the analysis in collaboration. Differences were solved by rereading, reanalysis and dialogue between LHH and JA. The analysis was thereafter discussed, reflected upon and refined with the author group. The interviews were conducted in Danish. After the analysis was conducted the used quotations were carefully translated into English by the authors to ensure the original meaning was maintained.

## Ethics

The North Denmark Region Ethics Committee approved the cohort study protocol (N-20180036). The additional qualitative protocol was forwarded to the Committee, although no approval was needed according to Danish law. All participants received written and oral information about the study before written consent was obtained. Participants were informed about confidentiality and anonymity, and the possibility of withdrawal of consent without consequences.

## Results

In total, 16 patients accepted the invitation to partake, and 11 were interviewed as three women had situations at home that made it inconvenient, and two women could not be reached. Participants were between 42 and 69 years of age, two participants had retired, one was unemployed, and the rest were employed full- or part-time (Table 2).

Data analysis identified five themes: (1) *Daily life was controlled and structured by pain;* (2) *The condition was acknowledged and taken seriously by treating professionals;* (3) *The participants´ experiences of the intervention–information is key;* (4) *Improved capability and autonomy in pain management* and (5) *The women´s perspectives on improving and expanding the intervention.* The themes were validated by quotations from the participants. Overall, the analysis showed that pain management was the central and most important intervention component as it enabled the women to minimize the impact on everyday life.

## Daily life was controlled and structured by pain

The women reported that their life was greatly influenced by pain caused by the lateral hip condition. Prior to participation in the study, they lacked knowledge of the condition, and those who had visited their general practitioner did not feel that they had received the help they needed.

**Table 2. Participant characteristics, N = 11.**

| ID | Location for the interview | Age at inclusion | Job situation | Symptom duration in months | Pain rating*: At the first control at hospital | Subjective degree of improvement in pain relief when interviewed# |
|---|---|---|---|---|---|---|
| 1 | Hospital | 59 | Unemployed | 120 | 8 | No improvement |
| 2 | Hospital | 68 | Retired | 7 | 7 | Full or nearly full recovery |
| 3 | Hospital | 55 | Full time | 13 | 3 | Considerable improvement |
| 4 | Hospital | 42 | Full time | 36 | 6 | Minor to moderate improvement |
| 5 | Hospital | 58 | Full time | 96 | 4 | Full or nearly full recovery |
| 6 | Hospital | 58 | Part time protected job | 48 | 7 | Considerable improvement |
| 7. | Hospital | 52 | Full time | 6 | 3 | Full or nearly full recovery |
| 8. | Hospital | 47 | Full time | 6 | 6 | Minor to moderate improvement |
| 9. | Own home | 58 | Full time | 36 | 5 | Minor to moderate improvement |
| 10. | Interview using the MS Teams teleconferencing platform | 45 | Full time | 12 | 6 | Considerable improvement |
| 11. | Own home | 69 | Retired | 24 | 5 | Full or nearly full recovery |

*The pain rating was the patients´ subjective usual pain on a Numeric Rating Scale (0–10 points), when entering the cohort study.

# The women were asked how they experienced their degree of improvement, and their response was categorized into one of four categories; no improvement, minor to moderate improvement, considerable improvement or full or nearly full recovery.

"Just the repeated times at the doctor saying;" Really, it keeps hurting, now I have tried this and have paused for a long time, but then it returns". In a way you kind of feel a little like a sissy and think"Okay, maybe it is just me.""

(ID4)

Therefore, many had resigned themselves to life with chronic pain and were thrilled to be included in a study targeting their situation. Their expectations upon partaking in the study were primarily to gain an understanding of what was wrong and, hopefully, experience considerable improvement.

The participants had experienced pain for a long time (between six months and ten years) and reported moderate to high levels of pain when it was at its worst. In describing the sensation of the pain, the informants used words as "murmuring", "deep", "heavy", "warm", "burning" and "radiating". The sensation did not originate from one specific spot but radiated from the greater trochanteric area.

*"It is murmuring and biting and radiating. When it was really bad, I felt it far down my legs. I get tired and feel weak in my legs. And then there is the pain in relation to movement, but also when I am sitting still."*

(ID7)

Although the women did not describe the characteristics of the pain identically, they consistently reported suffering from strong and persistent painful sensations over time. The pain was not experienced as a constant pain, rather it varied, changing according to their activity level and the specific position of the hip. One participant described the pain as being as painful as giving birth, while another could not go upstairs in the house when carrying things.

The condition was very painful and for many of the participants influenced almost every part of their daily lives, including sexual intimacy. The informants described that daily life was more cumbersome and that mundane tasks such as vacuuming and playing with their children or grandchildren was difficult. Sitting down or finding rest was also troublesome. This was also reflected in their work capability as the pain caused a need for changing positions, avoiding tasks and finding ways of managing throughout the working day. One woman described how this affected her:

*"I got afraid of not being capable of doing my job. Well, I went to my boss and got an agreement that I could lay down when needed, because I found it would be so embarrassing if somebody found me lying on a couch."*

(ID7)

Overall, their life situation was affected, and this had consequences for their well-being and surplus energy. One participant described her daily life as being controlled by the pain and if she tried to ignore the pain, it had consequences:

*"My everyday life was built around it (the pain) [. . .] I have always been known as the woman in high heels but that was not possible anymore. So. . .when I wore them, because I did, then I paid the price."*

(ID6)

The women structured everyday life and daily tasks to avoid further aggravation by stopping or avoiding aggravating activities. If they ignored the symptoms, they experienced pain for hours or days. In trying to manage the pain they also experienced other personal consequences:

*"I had also kind of lost my desire to dance. And I loved it so much. Because I felt I was somehow not included, as I many times had to say "no" to dance [. . .] I am not part of the group anymore, not in the same way, and this has been really hard."*

(ID2)

Although the majority expressed having the significant influence of lateral hip pain in their everyday life, not all the informants experienced severe pain during the day:

*"It has not bothered me. I have been able to attend to my work and I have also been able to run and perform physically as I'm used to [. . .] with me taking some Ibuprofen [. . .] which took away the pain."*

(ID5)

The pain affected not only the participants' waking hours, but also their sleep. Whilst some describe having difficulties falling asleep due to discomfort in their normal sleeping position, others woke up or slept intermittently.

Generally, the women used different strategies to manage the pain, and it became clear that the fear of pain controlled the participants´ daily life.

## The condition was acknowledged and taken seriously by treating professionals

Throughout all the interviews it was evident that after entering the cohort study, the women felt they were taken seriously. They noted that the two physiotherapists (CM, JEJ) not only acknowledged their condition, but were knowledgeable about it:

*"I feel I have been taken seriously and then it is easier for me to accept myself as well. I find I have become better at listening to my own body, because at the same time I feel I have been taken seriously and I have been examined and something has been done to treat it."*

(ID4)

The element of being taken seriously and acknowledged as an individual was essential to support the active process the participants started. Participants stated that both physiotherapists described what tests and treatments they would perform, and why, and participants reported that this was reassuring.

*"I think that JEJ has been so good to talk to. I was out there once a week. I could just ask him all the questions I wanted, and he has been good at answering them."*

(ID2)

This open approach made them feel safe and gave them confidence to act upon the information and advice provided. Further, participants appreciated the collaboration and alignment between the two physiotherapists. One called the experience "fantastic" and elaborated: *"The*

*collaboration. What CM said was in alignment with what JEJ did [. . .] you know, all the time there was alignment."* (ID6)

These alignment experiences were highly valued by the participants who felt encouraged and strived to internalize the guidance and advice. The participants expressed that they used it to continuously guide their daily activities during and after the intervention period. The character of the relationship was also emphasized:

*" Well, to me, it is the relational part that counts. Papers and pamphlets are fine as you then can go back [to them], but when it is presented by a real human being and one where you can feel there is knowledge behind, and it seems trustworthy and authentic . . .That means that I use it."*

(ID7)

The women valued the relationship with the physiotherapists and in many ways, it positively influenced their experiences, actions and motivation both during and after the intervention period.

## The participants´ experiences of the intervention–information is key

All participants experienced the intervention as informative and helpful regardless of the degree of improvement in symptoms at the time of the interview (See Table 1).

The examinations at baseline and follow ups, especially the clarification and explanation that nothing was wrong, were important elements for the women. Many had silently been worrying about their condition and whether it could be a severe condition e.g. cancer or hip arthrosis.

*"Well mentally, what has helped me is that CM both has scanned and taken measurements regarding the muscle strength, where the pain was situated and things like that [. . .] It has also mentally given confidence; that I know; Okay this is not something totally dangerous."*

(ID4)

Further, some found it motivating to follow their own development at the follow-ups:

*" It has been super to go to JEJ to see that it has become better and better. I can do more and more. And further, that it is not only my own measurement, but that I do the measurement together with him."*

(ID2)

Participants strongly valued the dialogues with the physiotherapists, along with the content of the advice and guidance pamphlets.

The practical advice was easy to handle and therefore easy to integrate in daily activities, although some habits could be difficult to change immediately. One participant expressed:

*"It is the thing about being attentive on what you do in your daily life. Really, the little things you do in your daily life, how big a significance it makes. This has been a huge eye-opener to me."*

(ID6)

Regarding experience of ESWT, the women expressed that the three treatments of ESWT caused strong pain, which seemed to be highest at the first treatment, after which they adapted a little to the sensation. In the aftermath of the ESWT the sensations and reactions differed, some felt something changed in their tissue whilst others felt the pain gradually diminishing. Seven of the women were convinced that ESWT was the significant element making the difference in relation to their pain:

*"I had a period of time where I didn´t feel any pain at all after these treatments and I was fine, but it is not totally gone yet [. . .] they (ESWT) have simply done something. I don´t know whether it loosens up or. . ."*

(ID8)

A few participants felt no relief from the ESWT and did not feel a distinct improvement:

*"I don´t think that the shockwaves helped a lot"*

(ID4)

The participants had different opinions and experiences regarding the exercises. Few performed all five exercises, some performed some of them regularly, others integrated the exercises in their fitness center program, and others only did them occasionally. Finally, participants avoided the exercises on days of increased pain:

*"I have some days, if I have been too hard on myself the day before and the pain is strong, then it goes without saying, that I should not do too many exercises. If it is really bad, then I skip the exercises."*

(ID1)

One exercise ("the clam" number 2, S1 Appendix) often provoked renewed pain. Participants found it impossible to perform without provoking strong pain that lasted for hours or days.

Overall, the examinations, advice and ESWT were predominantly experienced as positively contributing elements to the participants´ recovery, whereas the exercises resulted in more heterogeneous experiences.

## Improved capability and autonomy in pain management

Most participants became experienced at managing their pain during the intervention and evolved different strategies to cope with their symptoms long term. The most important mechanism for improvement for the patients did not seem to be knowledge and understanding of their GTPS condition but specifically their understanding of how to manage and cope with the pain. One element of achieving capability to manage the pain was the thorough examination, and the knowledge that this was not an uncommon condition. Further, it became evident that participants changed activity patterns following the intervention. A participant reported how she managed pain now if she had provoked it:

*"Then I sit down with a book and do not do any more that day [. . .] And I do follow the advice given; I still use it."*

(ID11)

Thus, the respect for the pain level and a graduated activity level was adapted in daily life activities. Another participant was asked if her pain level had changed:

*"The pain has completely changed character and I have learned to dose my activity level. I have learned to accept some pain and to reduce my activity or do something differently. And I feel calm knowing that I can handle it."*

(ID7)

She had learned to manage her level of physical activity and that was the mechanism that gave a feeling of being in control. Not all the participants consistently followed the advice:

*"There was one time where JEJ said, that maybe I went for too long walks. I have reduced it a little after that and that is actually the only advice I have listened to."*

(ID1)

Some participants were not as successful at managing their activity and pain, frequently coming close to or exceeding their pain threshold. Others did not feel the improvement they had hoped for but had still adapted to the recommended management strategies.

The women reported that the use of the strategies contributed to their adapting their lives and living better. The majority still experienced some degree of pain and consequences such as having to reduce activity levels, but overall, most women felt they now had the tools and felt capable to manage their pain.

### The women´s perspectives on improving and expanding the intervention

Even though the participants were predominantly satisfied with the intervention, many had suggestions for improvements, especially with respect to the exercises. These reflections are presented in Table 3 and validated by quotations.

Overall, the suggestions varied in relation to each participant´s state, needs and context, as illustrated in Table 3.

## Discussion

This qualitative study, nested within a cohort study that assessed the outcomes of a GTPS intervention, is the first to explore how women with GTPS manage daily life and how they experience the combined treatment of education, ESWT and exercise to understand the mechanisms possibly explaining their outcomes. The women described pain as the most severe problem related to GTPS and that pain took control over their daily life and identity, affecting physical, psychological and social domains. Pain education and management was a central element of the intervention for the women; as a measure of success was their ability to minimize pain and manage their daily life, during and after treatment. A mechanism supporting this was the therapeutic alliance between the physiotherapists and participants, as they felt taken seriously by the physiotherapists. Irrespective of the degree of improvement of their GTPS, the majority was satisfied with the intervention and experienced improvement. However, alternatives to the process of care delivery were suggested.

The women had been suffering from lateral hip pain for a long time and were clearly burdened. This is consistent with previous research [10, 15, 36]. Stephens et al. (2020) also showed that life was dominated by the pain and affected many domains in daily life [15]. Our study specifically showed how daily life was planned and structured around pain and this had

**Table 3. Ways of improving care from the participants´ perspective.**

| Suggestions | Quotation validation |
|---|---|
| **The intervention overall:** | |
| An overview of the treatment and time plan | *I have missed an overview of how many times ESWT and how many times to i.e JEJ\*.* (ID4) |
| What to do, if GTPS returns | *In reality, it would be nice with a wrapping up: "Now we are here, if it happens again, then this is the path to go."* (ID 3) |
| **The examination:** | |
| To have an explanation about the scanning results of the hip | *I haven´t seen them and I would actually like to have explained"what is happening in there?"* (ID9) |
| Role of imaging | *Personally, I think it was negative to have a scan, which only showed that my muscles and tendons were irritated, and that I knew already.*(ID7) |
| **The measurements:** | |
| Follow up on individual results | *Well, you probably have that overview, but I would have liked to know;" Here were you and here are you now", in numbers.* (ID3) |
| A possibility for repeated guidance | *I would **very much** like to have a follow-up in half a year [. . .] Now I work according to this and hope it goes well. But if it doesn´t, the questions I would have. . .* (ID6) |
| **Advice:** | |
| Information regarding pain and delayed pain reactions | *That is probably to tell them, that the pain can arrive the day after and two days after.* (ID1) |
| **Exercises:** | |
| Group-based exercise training | *Well, I find that group- based is good, even though it is with individual exercises; that you show up and show up at a specific time point.* (ID5) |
| Group-based exercise training on prescription (training paid by the Danish Government) | *You should have a"green prescription" telling that you should attend group-based training (paid by government) [. . .] You know, tell people it is a part of a medication.* (ID4) |
| Continuous guidance | *JEJ\* knew it all and I did do them (the exercises) out there. But you know, then you still can have doubts once in a while,"what was it exactly here, should I do more?"* (ID8) |
| Fitness-centre adapted training | *That there was someone who was educated to train with us, so that you had a fitness center.* (ID6) |
| More than one instruction in exercises | *We did go through it and at that time I could do it. Even though it would have been irritating to drive to the city again, I would have preferred that we had practiced the exercises together again.* (ID9) |
| **Shock wave therapy treatment:** | |
| Additional ESWT treatments | *A possibility of having a couple of treatments more, as I already the first time could feel, how much it helped me.* (ID8) |
| A written description of dosage and frequency of ESWT, so that a local physiotherapist can take over the ESWT | *To know exactly what the treatment is called and what they have done. A description like that I could bring to my own physiotherapist.* (ID9) |
| Information/dissemination in general: | |
| Dissemination to general practitioners and public | *This is actually why you do a project like this: to publish that you have tried this. So that it becomes visible for others as well.* (ID2) |
| Facebook group for people with GTPS for sharing experiences | *It could be an idea to start a Facebook group, because there could then be some experience- sharing. Maybe some found out they would train together or other things. . .* (ID10) |

JEJ\*: The physiotherapist at the private clinic.

consequences for their quality of life and different aspects of daily life, e.g. walking even short distances, sitting for a longer period or working for hours were challenging. Previous studies report that GTPS pain affected work, physical activity and quality of life [13, 32, 37]. This was also the case for participants in this study. Therefore, from the women´s perspective the pain severity, duration and tools to self-manage were key elements to the intervention success. Such elements should be addressed early in the clinical care as the vast majority had experienced GTPS pain for years. However, participants did not feel that their physicians had met their needs. This issue does not seem to be specific for women with GTPS as recent qualitative studies exploring other musculo-skeletal conditions also describe similar experiences in their target populations [38–40]. But the women in this study also stressed that they felt they had been taken seriously by the physiotherapists which may imply that they had not previously felt taken seriously. However, we did not explicitly ask if their physician had taken them seriously and future research should look into this.

The therapeutic relationship and subsequent alliance were experienced as positive, strong, reassuring and important by the participants. A review by Manzoni et al. (2018) regarding therapeutic alliance and pain relief in relation to physiotherapy treatment for musculoskeletal injuries showed no apparent link between the two [41]. Our findings indicate that a focus solely on pain relief may be too narrow. In our study, the therapeutic alliance provided reassurance and confidence to the participants' understanding and managing their individual pain patterns. These concepts were key to the participants´ insight about the condition as manageable, with most participants becoming motivated to make an effort to learn how to manage the pain instead of being disappointed that it had not completely disappeared. The therapeutic alliance, characterized by acknowledgement of the individuals' situation, addressing individuals' needs, and a secure therapeutic relationship, may be the mechanisms that supported the development of intrinsic motivation and self-efficacy as described by Ryan and Deci [42]. These findings are supported by a scoping review indicating that an enhanced therapeutic alliance may contribute to intervention adherence in musculoskeletal physiotherapy interventions [26]. The women seemingly had achieved a quite high pain self-efficacy due to the intervention elements, where the therapeutic alliance seemed to be of particular importance for their learning. Both physiotherapists in our intervention were experienced specialists in managing GTPS, and the effects may not be directly transferable to other settings where the treating physiotherapists are not specialists in the management of this condition.

The women were concerned about their pain condition, especially wanting to be reassured that nothing serious was wrong with their hip joint such as cancer, severe hip arthritis or the need for surgery. This reassurance seemed to be a key mechanism to their management and recovery. Unlike in Stephens et al. [15] our participants did not express confusion regarding the diagnostic labels or patho-anatomic explanations given. Rather, the women in our study strived to find ways of handling their pain condition when reassured that nothing was seriously wrong. The most important mechanism for improvement therefore did not seem to be the knowledge and understanding of the GTPS condition, but their learning and understanding of how to manage and cope with the pain in daily life. However, as other studies also suggest [15, 20], the condition should be viewed as complex and multi-dimensional and addressed accordingly.

Most participants felt that ESWT had a positive effect. It seemed that the ESWT was the trigger that encouraged them to strive for self-management and control. These findings were not fully supported in a recent qualitative study exploring the subjective experiences of ESWT to patients with tendinopathies in general [20]. Leung and colleagues found that participants felt that self-management measures were equally or more important than ESWT to help treat their tendinopathies [20]; elements which the participants in this study also found important.

The participants described mixed experiences of the exercises in the intervention, and compliance to exercises were mixed. While the pamphlet and the advice provided during the intervention were useful and appreciated by the women, the implementation of the specific exercises caused a recurrence of pain for some participants. A few integrated them in their normal schedule without any side effects, whereas some did not do them very often. The lack of compliance may have been due to an increase in pain related to undertaking the exercises, or the lack of interest for the specific exercise regime. The issues related to non-adherence and low compliance is well-known in general [43] and in physiotherapy as well regarding non-supervised exercises [44, 45] and should be addressed carefully, i.e by a thorough dialogue with the women regarding the type and intensity of pain being acceptable during exercise as well as carefully selecting which and how many exercises to provide to each woman.

Our findings suggest that women with GTPS prefer, and possibly learn and respond best, to an individualized intervention. Exercise and education on how to reduce load in relation to the hip have demonstrated to be effective in two RCTs [10, 18], however women in our study emphasized the value of ESWT and their relationship with the physiotherapists as well. Further, it was demonstrated by the quotes of the women that an individualized approach to the prescribed exercises, as well as the number exercise sessions, ESWTs and maybe even assessments at the hospital should be negotiable in relation to their individual needs. An individualized approach seems necessary to accommodate the needs and options for each woman as there was not one clear approach suggested as optimal to manage the condition.

The course of treatment and the way of including the women with lateral hip pain in the treatment was primarily experienced as positive and meaningful. Therefore, the approach, encompassing a few alterations, could be considered a framework in future trials and treatment strategies. The framework should address the following key elements when developing an individualized approach: i) Emphasis on GTPS as treatable and not severe arthritis or cancer, when explaining the diagnosis, ii) Emphasis on the importance of taking the patients seriously and acknowledging their condition when building up a therapeutic relationship, iii) Alignment in the messages and explanations between health professionals, iv) Emphasis on the learning process to self-managing pain in daily life, instead of a focus solely on recovery, and finally iv) Emphasis in relation to education, ESWT and exercises, that one size does not fit all and individual preferences should be prioritised.

### Methodological considerations

The participants met the predefined criteria of variation in age, symptoms and duration. The analytical process was carried out by two authors in the initial phase, first independently and thereafter in collaboration. Subsequently the author group was involved to ensure analytical sensitivity and agreement. This process was performed and presuppositions were reflected upon to achieve trustworthy and credible findings [34]. This rigorous analytical process was considered a strength of the study. Further, the authors included both non-clinical and clinical researchers, thereby possessing knowledge within the clinical as well as the theoretical and methodological field. The 11 interviews provided rich data and views, however, as this was a rather small sample, saturation cannot be guaranteed, neither can it be guaranteed that all viewpoints were expressed. Braun and Clarke (2013) state that the term saturation invokes a more positivist model of qualitative research and the researcher should consider which approach should be taken [34]. Recent papers by Saunders et al. (2018) and Thorne (2020) discuss different approaches to saturation [46, 47]. Thorne problematizes the term and states that it is more important to assess whether there is sufficient depth, richness and coherence within the reported findings to make interpretations and conclusions relevant and credible in relation

to complex clinical phenomena. We find the theoretical approach in this study useful as the analysis of the data using this approach helped us to explore and emphasize that the women individually constructed their understanding and management of the GTPS condition in their given contexts. We find that our data and analysis have provided depth, richness and coherence within the findings and that the interpretations and conclusions are relevant and credible in relation to complex clinical field of GTPS. It may be considered a limitation that no interviewees were presented with the results of this qualitative study although this could have validated the findings. A further limitation of the study was that the therapeutic relationship may have been influenced by the context of being in a research study and does therefore not fully mimic a real-life situation and the physiotherapists providing the treatment were specialized within musculo-skeletal conditions. Finally, it should be noticed that the women have experienced symptoms of varying duration and varying pain. Therefore, transferring the findings to other contexts should be done cautiously [34].

## Conclusion

This qualitative study, which was nested within a cohort study assessing the outcomes of a treatment intervention, is the first to explore how women with GTPS manage daily life and how they experience the combined treatment of education, ESWT and exercise. Pain was the most severe problem for the women and affected them physically, psychologically and socially. Learning to manage the pain was the most important element for the women so as to be able to minimize pain and manage their daily life. Most participants were satisfied with the intervention and experienced improvements. However, they also suggested alterations to the intervention. These new findings regarding how women with GTPS experienced and managed pain in daily life, and how they experience and adapt to treatment elements, are important additions to the current evidence to inform future clinical practice for women with GTPS alongside evidence of effectiveness and clinicians' clinical reasoning. This new knowledge may be used to inform individualized patient education, treatment and evaluation strategies for women with the painful and debilitating condition of GTPS.

## Supporting information

**S1 Appendix. Home-exercises provided for the participants.**
(TIF)

## Author Contributions

**Conceptualization:** Jane Andreasen, Angela Fearon, Dylan Morissey, Jens Erik Jorgensen, Carsten M. Mølgaard.

**Formal analysis:** Jane Andreasen, Laura H. Hjørnholm.

**Investigation:** Jane Andreasen, Laura H. Hjørnholm.

**Methodology:** Jane Andreasen, Angela Fearon, Dylan Morissey, Jens Erik Jorgensen, Carsten M. Mølgaard.

**Writing – original draft:** Jane Andreasen.

**Writing – review & editing:** Jane Andreasen, Angela Fearon, Dylan Morissey, Laura H. Hjørnholm, Jens Kristinsson, Jens Erik Jorgensen, Carsten M. Mølgaard.

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
