## [Decision Letter · Decision Letter 0]

1 Nov 2021

PONE-D-21-27762"I feel I have been taken seriously" Women’s experience of greater trochanteric pain syndrome treatment - a nested qualitative studyPLOS ONE

Dear Dr Andreasen. ,

Thank you for submitting your manuscript to PLOS ONE. After careful consideration, we feel that it has merit but does not fully meet PLOS ONE’s publication criteria as it currently stands. Therefore, we invite you to submit a revised version of the manuscript that addresses the points raised during the review process.

Please address the methodological issues raised by the reviewer.

Please submit your revised manuscript by  Dec 16 2021 11:59PM. If you will need more time than this to complete your revisions, please reply to this message or contact the journal office at plosone@plos.org. Please include the following items when submitting your revised manuscript:A rebuttal letter that responds to each point raised by the academic editor and reviewer(s). You should upload this letter as a separate file labeled 'Response to Reviewers'.A marked-up copy of your manuscript that highlights changes made to the original version. You should upload this as a separate file labeled 'Revised Manuscript with Track Changes'.An unmarked version of your revised paper without tracked changes. You should upload this as a separate file labeled 'Manuscript'.If applicable, we recommend that you deposit your laboratory protocols in protocols.io to enhance the reproducibility of your results. Protocols.io assigns your protocol its own identifier (DOI) so that it can be cited independently in the future. For instructions see: https://journals.plos.org/plosone/s/submission-guidelines#loc-laboratory-protocols. Additionally, PLOS ONE offers an option for publishing peer-reviewed Lab Protocol articles, which describe protocols hosted on protocols.io. Read more information on sharing protocols at https://plos.org/protocols?utm_medium=editorial-email&utm_source=authorletters&utm_campaign=protocols.

We look forward to receiving your revised manuscript.

Kind regards,

Rosemary Frey

Academic Editor

PLOS ONE

Journal Requirements:

2. Please amend your Methods section and Ethics Statement to state what form of consent was provided by participants - i.e., written, verbal, etc.

Reviewers' comments:

Reviewer's Responses to Questions

**Comments to the Author**

1. Is the manuscript technically sound, and do the data support the conclusions?

Reviewer #1: Yes

2. Has the statistical analysis been performed appropriately and rigorously? 

Reviewer #1: Yes

3. Have the authors made all data underlying the findings in their manuscript fully available?

Reviewer #1: Yes

4. Is the manuscript presented in an intelligible fashion and written in standard English?

Reviewer #1: Yes

5. Review Comments to the Author

Reviewer #1: Thank you to the authors for this study which continues to add to the development of understanding of this condition. Some feedback is provided to improve the clarity of the paper

Abstract

Pg 2, line 22-23 Suggesting adding in ‘for GTPS’ at the end of this sentence

Pg 2, lines 36-37, review this wording for clarity ‘ important element of the programme to the women to be able….’

Introduction

Pg 3, line 62 add supporting references for the RCTs and qualitative studies

Pg 4, line 67 add detail of the sample in the qualitative study e.g. number in study, men and women

Pg 4, line 68 review this wording for clarity ‘was seen as or more important than…’

Pg 4, line 77 suggest alternative clearer wording for ‘improve their situation’

Methods

Pg 4, line 88 what is the specialized physiotherapist specialised in?

Also clarify in relation to the second specialist physio (JEJ)

Pg 5, line 95 what does a ‘thorough introduction mean and entail?

Pg 5, line 10 change ‘Those that’ to ‘Those who’

Pg 5, line 107, can you account for the 5 who agreed to take part, but were not interviewed? Why were they not interviewed?

Pg 5, lines 97-98 suggesting moving the sentence on COREQ guidelines to the start of the methods section.

Pg 6, Table 1- suggest moving Table 1 participant characteristics to the results section.

In column 1 suggest changing heading to ‘location of the interview’

In row 10, is this MS Teams? Suggest being clearer e.g. ‘MS Teams teleconferencing platform’

Pg 6, line 118 add supporting references for the literature on which you based the topic guide

Pg 7, line 122 – does the positive test refer to a positive COVID test?

Pg 7, Table 2: what does ‘private domain’ mean, ‘controls and follow-ups’ also not clear.

Were interviews conducted in the Danish or English language. In relation to the quotations used, were these then translated into English and by whom (how can you ensure that information was not lost in translation?)

Pg 10, line 177 ‘can you clarify what you mean by intimacy- is this referring to sexual intimacy. I would suggest being clearer about this as an important topic.

Results

Pg 13, line 239, the title of this theme is not as clear as the other 3. I wonder if it is possible to elaborate the title to increase clarity?

Pg 13, line 254, I think ‘alfa’ should be spelt ‘alpha’

Pg 14, line 264- how many re you referring to in relation to ‘a substantial number of women’

Pg 7, Table 3. The wording of the items in the first column are not fully clear. Were these ‘themes’ devised by the researchers. Examples which I suggest require some rewording include

• ‘A plan over the treatment element and time schedule

• Plan, if it comes back

• Is a scan necessary ( is this a question?) suggest change to ‘role of imaging’?

• Follow-up on results ( does this refer to individual results or overall study results?)

• What is the difference between ‘group-based exercise training’ and ‘group-based exercise training on prescription’?

• Typing errors on ‘continuous guidance’ – again, not fully clear what this means, compared to the other headings

• Dosage, frequency and intention explicated (not sure what this word means or even exists?)

• Information or dissimilation (not sure what this word means?)

Facebook group – context not clear

Discussion

Pg, 19, in the first line here, it is important to clarify that the experience of GTPS was in the context of a research study ( ie the cohort study), as many of the findings may relate to the effect of being in the study e.g. Hawthorne effect

Pg 20, the therapeutic relationship also may have been influenced by the context of being in a research study and this is worth acknowledging as a limitation of the study, as it does not mimic real life

Pg 20, line 355 review the wording ‘the competent meeting of the individuals needs…’ to enhance clarity

Pg 21, line 362 suggesting changing ‘in the field of GTPS’ to ‘ in the management of GTPS’. In relation to the effects not being transferable to other settings, clarify if this is because is was nested in a cohort study or because of the specialist physiotherapists ( in reality, it could be both).

Pg 21, line 374, in relation to ESWT, do you mean it had a positive effect- please clarify

Pg 21, line 377, add a supporting reference in relation to the qualitative study.

Pg 21, line 377 is the word ‘herem’ a typing error’, please correct.

Pg 21, line 381, review wording ‘dialogues with advice’ for clarity

Pg 22, line 392, change the wording ‘relationship to’ to ‘relationship with’

Pg 22, line 398 what do you mean by ‘the way of approaching women’ – does this refer to recruitment? Please clarify

Pg 22, line 404, it is not surprising that there was alignment between the two physiotherapists’ messaging to patients it this was part of the study. Was there any exploration of the information/explanations received from other health professionals- this is a key difference between this and Stephen’s study and should be acknowledged.

Pg 22, line 416, in relation to saturation, please clarify what you mean. How would you have determined data saturation? Suggest reviewing and referring to recent papers in relation to the concepts/controversies around data saturation.

Pg 23, Please present study limitations more clearly in this section. The heading methodological considerations is not clear. Perhaps changes to study strengths and limitation?

Pg 23, line 417 do you mean the results of this qualitative study or the results of the cohort study, please clarify

The conclusion should explicitly state that the experience of women in this study is nested within a cohort study

Pg 23, review wording ‘are important knowledge…’ to enhance clarity

6. PLOS authors have the option to publish the peer review history of their article (what does this mean?). If published, this will include your full peer review and any attached files.

Reviewer #1: **Yes: **Helen French

---

## [Author Response · Author response to Decision Letter 0]

7 Feb 2022

Response letter to Academic editor Rosemary Frey,

Dear Rosemary Frey,

Thank you very much for the constructive comments from Dr. Helen French and for the opportunity to revise and improve the manuscript. As recommended, we have responded to each point raised by the reviewer before uploading a 'Revised Manuscript with Track Changes' as well as a manuscript without track changes labeled 'Manuscript'. All authors have approved the final version of this revised manuscript.

We have addressed each point raised by the reviewer separately with the following structure:

1. Reviewer comment

2. Author response

3. Author actions

Comments to the Author

1. Is the manuscript technically sound, and do the data support the conclusions?

Reviewer #1: Yes

2. Has the statistical analysis been performed appropriately and rigorously? 

Reviewer #1: Yes

3. Have the authors made all data underlying the findings in their manuscript fully available?

Reviewer #1: Yes

4. Is the manuscript presented in an intelligible fashion and written in standard English?

Reviewer #1: Yes

5. Review Comments to the Author

Reviewer #1: Thank you to the authors for this study which continues to add to the development of understanding of this condition. Some feedback is provided to improve the clarity of the paper

Authors: Thank you very much for your time and constructive comments.

Abstract

Pg 2, line 22-23 Suggesting adding in ‘for GTPS’ at the end of this sentence

Authors: Agree

Changes to text: Education, extracorporeal shockwave therapy (ESWT) and exercise have good group level evidence for efficacy in clinical trials and are increasingly used in routine care for patients with greater trochanteric pain syndrome (GTPS).

Pg 2, lines 36-37, review this wording for clarity ‘ important element of the programme to the women to be able….

Authors: We have reviewed and changed the wording

Changes to text: Learning how to manage pain was experienced as the most important element of the program to the women to be able to minimize pain and manage daily life.

Introduction

Pg 3, line 62 add supporting references for the RCTs and qualitative studies

Authors: We have added the supporting references.

Changes to text: Previous assumptions about GTPS being a mild, self-limiting condition (16,17) have therefore been disproven and the marked absence of research on interventions and experience is being addressed, as shown by a recent acceleration in the number of randomised controlled trials and qualitative studies (10,15,18–22).

Pg 4, line 67 add detail of the sample in the qualitative study e.g. number in study, men and women

Authors: We have added details of the sample.

Changes to text: A qualitative study including eleven participants (eight males and three females) found that the participants with tendinopathies were largely unaware of the mechanism of action in ESWT and that education was seen as or more important than ESWT

therapy (21).

Pg 4, line 68 review this wording for clarity ‘was seen as or more important than…’

Authors: We have revised the wording.

Changes to text: A qualitative study including eleven participants (eight males and three females) found that the participants with tendinopathies were largely unaware of the mechanism of action in ESWT and that education in how to handle their condition was seen as important or even more important than ESWT therapy (21).

Pg 4, line 77 suggest alternative clearer wording for ‘improve their situation’

Authors: We have rephrased the sentence for clarity.

Changes to text: Recent work carried out to inform trial design has begun to explore participants’ understanding of GTPS, their pain and their beliefs regarding activity and exercise (15) but we need to better understand how they experience their daily life with GTPS, how participation in a multi-modal treatment program is experienced by the women, and how it may improve their situation regarding pain and functional level.

Methods

Pg 4, line 88 what is the specialized physiotherapist specialised in?

Also clarify in relation to the second specialist physio (JEJ)

Authors: We have clarified in relation to both physiotherapists.

Changes to text:… by a physiotherapist specialized in musculo-skeletal physiotherapy (CM) at baseline, and after 12 and 26 weeks.

Changes to text: The ESWT, exercise therapy and additional education and advice was then performed at a physiotherapy clinic with a second physiotherapist specialized in musculo-skeletal physiotherapy (JEJ) .

¬¬¬¬¬¬¬¬¬¬¬¬¬___________________

Pg 5, line 95 what does a ‘thorough introduction mean and entail?

Authors:

Changes to text: The thorough introduction consisted of an explanation to the findings and diagnose of the individual participant at inclusion by CM. During the first session by JEJ, which included the ESWT intervention, the participants were once more informed orally and in writing about the known mechanisms of how GTPS and the pain experienced by GTPS may be managed in everyday life. The education provided was to educate the participants to avoid movements and situations causing pain (10,28). The education was repeated and targeted each patient´s experience, during the following two treatment sessions. At the six-week follow up, participant experience was addressed, and any positive or negative pain experiences were used to facilitate further pain management strategies specific for the individual participant. Thereby moving from a “general” education towards a customized “personal” education specific to the individual participant´s daily context and way of life.

Pg 5, line 101 change ‘Those that’ to ‘Those who’

Authors: We have changed the wording.

Changes to text: Those who consented were invited to interview in accordance with the purposive

sampling strategy, designed to ensure participants of different demographic characteristics, age,

severity and chronicity of GTPS were represented in order to elicit nuances and sufficient richness of data (23).

Pg 5, line 107, can you account for the 5 who agreed to take part, but were not interviewed? Why were they not interviewed?

Authors: Yes, we have added this to the manuscript and moved it to the results section.

Changes to text: In total, 16 patients accepted the invitation to partake, and 11 were interviewed as three women had situations at home that made it inconvenient, and two women could not be reached.

Pg 5, lines 97-98 suggesting moving the sentence on COREQ guidelines to the start of the methods section.

Authors: We have moved the sentence to the start of the methods section.

Changes to text: This report was guided by the Consolidated Criteria for Reporting Qualitative Research (27). This qualitative study was nested within a cohort study (N=60) conducted in an outpatient hospital clinic and a physiotherapy clinic in Denmark.

Pg 6, Table 1- suggest moving Table 1 participant characteristics to the results section.

In column 1 suggest changing heading to ‘location of the interview’

Authors: We have moved Table 1 to the results section and changed the wording in column 1.

Changes to text: Results: In total, 16 patients accepted the invitation to partake, and 11 were interviewed. Participants were between 42 and 69 years of age, two participants had retired, one was unemployed, and the rest were employed full- or part-time (Table 1).

Column 1: Location for the interview.

In row 10, is this MS Teams? Suggest being clearer e.g. ‘MS Teams teleconferencing platform’

Authors: We have clarified.

Changes to text: Interview using the MS Teams teleconferencing platform.

Pg 6, line 118 add supporting references for the literature on which you based the topic guide

Authors: We have added references

Changes to text: The topic guide for the semi-structured interviews (Table 2) was based on research literature and clinical experience of the research team (10,13,14,20,31,32).

Pg 7, line 122 – does the positive test refer to a positive COVID test?

Authors: Yes, we have clarified.

Changes to text: During the COVID-19 pandemic restrictions, one interview was performed using MS Teams teleconferencing platform due to a positive COVID-19 test in the interviewee’s family.

Pg 7, Table 2: what does ‘private domain’ mean, ‘controls and follow-ups’ also not clear.

Authors: we have altered and clarified the wording

Changes to text: Everyday life with GTPS; the experiences related to work life, private life and leisure time before entering the cohort study.

Changes to text: Experiences and reflections upon their participation in the different elements of the cohort study:

• At the baseline assessment and at the follow-up assessments in the cohort study.

Were interviews conducted in the Danish or English language. In relation to the quotations used, were these then translated into English and by whom (how can you ensure that information was not lost in translation?)

Authors: We agree this is important to clarify. The author group consists of four researchers with mother- tongue Danish and skilled in English, two researchers with mother-tongue English/Australian and one with bilingual Danish-English tongue. We have inserted a sentence in relation the interviews performed in Danish and one in relation to the translation of the quotations- in the end of the data analysis section. 

Changes to text: The interviews were conducted in Danish. After the analysis was conducted the used quotations were carefully translated into English by the authors to ensure the original meaning was maintained.

Pg 10, line 177 ‘can you clarify what you mean by intimacy- is this referring to sexual intimacy. I would suggest being clearer about this as an important topic.

Authors: Thank you, we agree and have clarified.

Changes to text: As the quotation illustrates, the condition was very painful and for many of the participants influenced almost every part of their daily lives, including sexual intimacy.

Results

Pg 13, line 239, the title of this theme is not as clear as the other 3. I wonder if it is possible to elaborate the title to increase clarity?

Authors: It is possible to change for clarity of the content. The theme “Participants´ experienced outcomes” has been changed to “The participants´ experiences of the intervention”

Changes to text: “The participants´ experiences of the intervention”

Pg 13, line 254, I think ‘alfa’ should be spelt ‘alpha’

Authors: Thank you, we have changed the spelling to ”alpha”.

Changes to text: “I have used them 100 %, I would say. Many of the elements, the sleeping on the back, that is alpha and omega for me, and I will never stop that.” (ID10)

¬¬¬¬¬¬¬¬¬¬¬¬¬¬¬¬¬

Pg 14, line 264- how many are you referring to in relation to ‘a substantial number of women’

Authors: We refer to 7 women stating this. 

Changes to text: Seven of the women were convinced that ESWT was the significant element making the difference.

Pg 7, Table 3. The wording of the items in the first column are not fully clear. Were these ‘themes’ devised by the researchers. Examples which I suggest require some rewording include

• ‘A plan over the treatment element and time schedule

• Plan, if it comes back

• Is a scan necessary ( is this a question?) suggest change to ‘role of imaging’?

• Follow-up on results ( does this refer to individual results or overall study results?)

• What is the difference between ‘group-based exercise training’ and ‘group-based exercise training on prescription’?

• Typing errors on ‘continuous guidance’ – again, not fully clear what this means, compared to the other headings

• Dosage, frequency and intention explicated (not sure what this word means or even exists?)

• Information or dissimilation (not sure what this word means?)

* Facebook group – context not clear

Authors: Thank you for the suggestions and comments in relation to Table 3. The “themes” were devised of the suggestions given by the women. We have clarified the wording of the items in the first column. 

The difference between ‘group-based exercise training’ and ‘group-based exercise training on prescription’ is that if it is on prescription it would be free of charge for the participants over time and that it is prescribed just like medications from the physician.

Changes to text in the first row of Table 3:

• An overview of the treatment and time plan 

• What to do, if GTPS returns

• Role of imaging

• Follow-up on individual results 

• A possibility for repeated guidance 

• A written description of dosage and frequency of ESWT so that a local physiotherapist can take over the ESWT 

• Information or dissemination 

• Facebook group for people with GTPS for sharing experiences

Discussion

Pg, 19, in the first line here, it is important to clarify that the experience of GTPS was in the context of a research study ( ie the cohort study), as many of the findings may relate to the effect of being in the study e.g. Hawthorne effect

Authors: We agree, thank you for pointing this out.

Changes to text: This qualitative study, nested within a cohort study that assessed the outcomes of a GTPS intervention, is the first to explore how women with GTPS manage daily life and how they experience the combined treatment of education, ESWT and exercise to understand the mechanisms possibly explaining their outcomes. 

Pg 20, the therapeutic relationship also may have been influenced by the context of being in a research study and this is worth acknowledging as a limitation of the study, as it does not mimic real life

Authors: Again, we agree on this point as well. We have added this as a limitation of the study.

Changes to text: A further limitation of the study was that the therapeutic relationship may have been influenced by the context of being in a research study and does therefore not mimic a real-life situation and the physiotherapists providing the treatment were specialized within musculo-skeletal conditions.

Pg 20, line 355 review the wording ‘the competent meeting of the individuals´ needs…’ to enhance clarity

Authors: We have used a more precise wording.

Changes to text: the professional meeting of the individuals´ needs.

Pg 21, line 362 suggesting changing ‘in the field of GTPS’ to ‘ in the management of GTPS’. In relation to the effects not being transferable to other settings, clarify if this is because is was nested in a cohort study or because of the specialist physiotherapists ( in reality, it could be both).

Author: We agree, and we have clarified what was the message in this sentence.

Changes to text: Both physiotherapists in our intervention were experienced specialists within the management of GTPS, and the effects may not be directly transferable to other settings where the treating physiotherapists are not specialists within the condition. 

Pg 21, line 374, in relation to ESWT, do you mean it had a positive effect- please clarify

Authors: We have clarified.

Changes to text: Most women felt that ESWT had a positive effect.

Pg 21, line 377, add a supporting reference in relation to the qualitative study.

Authors: We have added the reference.

Changes to text: These findings were not fully supported in a recent qualitative study exploring the subjective experiences of ESWT to patients with tendinopathies in general (20).

Pg 21, line 377 is the word ‘herem’ a typing error’, please correct.

Authors: We have reworded.

Changes to text: Leung and colleagues found that participants felt that self-management measures were equally or more important than ESWT to help treat their tendinopathies (20); elements which the participants in this study also found important.

Pg 21, line 381, review wording ‘dialogues with advice’ for clarity

Authors: We Have reviewed and changed for clarity.

Changes to text: While the pamphlet and the advice provided during the intervention were useful and appreciated by the women, the implementation of the specific exercises caused a recurrence of pain for some participants.

¬¬¬¬¬¬¬¬¬¬¬¬¬¬_____________________

Pg 22, line 392, change the wording ‘relationship to’ to ‘relationship with’

Authors: Thank you.

Changes to text: …..however, women in our study emphasized the value of ESWT and their relationship with the physiotherapists as well.

Pg 22, line 398 what do you mean by ‘the way of approaching women’ – does this refer to recruitment? Please clarify

Authors: We have clarified.

Changes to text: The course of treatment and the way of including the women with lateral hip pain in the treatment was primarily experienced as positive and meaningful.

Pg 22, line 404, it is not surprising that there was alignment between the two physiotherapists’ messaging to patients it this was part of the study. Was there any exploration of the information/explanations received from other health professionals- this is a key difference between this and Stephen’s study and should be acknowledged.

Authors: Yes, some women described experiences where they had consulted a doctor, but had not received the help (added as a quotation to the results section page 10 line 171-173) they needed (page 9 line 169-170), so that is why we have the second and third key element included as recommendations for future practice. We have elaborated a little further on this as we find there may be an issue specifically present for women with GTPS as they specifically stressed that they felt taken seriously by the physiotherapists. We have added a quotation in the result section in the first theme illustrating that the women did not feel their needs were met at the physician and discussed this more in the discussion section.

Changes to text: 

Results section: ”Just the repeated times at the doctor saying; ” Really, it keeps hurting, now I have tried this and have paused for a long time, but then it returns”. In a way you kind of feel a little like a sissy and think ”Okay, maybe it is just me”” ID4

Discussion section: However, participants did not feel that their physicians had met their needs. This issue does not seem to be specific for women with GTPS as recent qualitative studies exploring other musculo-skeletal conditions also describe similar experiences in their target populations (38–40). But the women in this study also stressed that they felt they had been taken seriously by the physiotherapists which may imply that they had not previously felt taken seriously. However, we did not explicitly ask if their physician had taken them seriously and future research should look into this. 

Pg 22, line 416, in relation to saturation, please clarify what you mean. How would you have determined data saturation? Suggest reviewing and referring to recent papers in relation to the concepts/controversies around data saturation.

Authors: We have looked further into the literature and especially the recent controversies regarding saturation in qualitative research that has been discussed, both regarding the many different ways of defining it and whether it is a legitimate argument or an artificial and constructed argument for stopping the data collection (Saunders 2018, Thorne 2020). Braun and Clarke (2013) state that the term saturation invokes a more positivist model of qualitative research and the researcher should consider which approach should be taken (Brown and Clarke 2013). Recent papers by Saunders et al. (2018) and Thorne (2020) discuss different approaches to saturation and especially Thorne problematizes the term and states that it is more important to assess whether there is sufficient depth, richness, detail and coherence within the reported findings to make interpretations and conclusions relevant and credible in relation to complex clinical phenomena. We find that our data and analysis have provided depth, richness and coherence within the findings and that the interpretations and conclusions are relevant and credible in relation to complex clinical field of GTPS. Therefore, we have looked more into data richness and depth, and coherence in the findings.

Changes to text: Braun and Clarke (2013) state that the term saturation invokes a more positivist model of qualitative research and the researcher should consider which approach should be taken (Brown and Clarke 2013). Recent papers by Saunders et al. (2018) and Thorne (2020) discuss different approaches to saturation (Saunders, Thorne). Especially Thorne problematizes the term and states that it is more important to assess whether there is sufficient depth, richness and coherence within the reported findings to make interpretations and conclusions relevant and credible in relation to complex clinical phenomena. We find that our data and analysis have provided depth, richness and coherence within the findings and that the interpretations and conclusions are relevant and credible in relation to complex clinical field of GTPS. 

Pg 23, Please present study limitations more clearly in this section. The heading methodological considerations is not clear. Perhaps changes to study strengths and limitation?

Authors: We have changed the heading and presented the strengths and limitations more clearly.

Changes to text: 

Study strengths and limitations.

The participants met the predefined criteria of variation in age, symptoms and duration. The analytical process was carried out by two authors in the initial phase, first independently and thereafter in collaboration. Subsequently the author group was involved to ensure analytical sensitivity and agreement. This process was performed and presuppositions were reflected upon to achieve trustworthy and credible findings (34). This rigorous analytical process was considered a strength of the study. Further, the authors included both non-clinical and clinical researchers, thereby possessing knowledge within the clinical as well as the theoretical and methodological field. The 11 interviews provided rich data and views, however, as this was a rather small sample, saturation cannot be guaranteed, neither can it be guaranteed that all viewpoints were expressed. Braun and Clarke (2013) state that the term saturation invokes a more positivist model of qualitative research and the researcher should consider which approach should be taken (34). Recent papers by Saunders et al. (2018) and Thorne (2020) discuss different approaches to saturation (43,44). Especially Thorne problematizes the term and states that it is more important to assess whether there is sufficient depth, richness and coherence within the reported findings to make interpretations and conclusions relevant and credible in relation to complex clinical phenomena. We find that our data and analysis have provided depth, richness and coherence within the findings and that the interpretations and conclusions are relevant and credible in relation to complex clinical field of GTPS. It may be considered a limitation that no interviewees were presented with the results of this qualitative study although this could have validated the findings. A further limitation of the study was that the therapeutic relationship may have been influenced by the context of being in a research study and does therefore not mimic a real-life situation and the physiotherapists providing the treatment were specialized within musculo-skeletal conditions. Therefore, transferring the findings to other contexts should be done cautiously (34). 

Pg 23, line 417 do you mean the results of this qualitative study or the results of the cohort study, please clarify

Authors: We have clarified.

Changes to text: No interviewees were presented with the results of this qualitative study, although this could have validated the findings.

The conclusion should explicitly state that the experience of women in this study is nested within a cohort study

Authors: We agree, and we have added a sentence in the start of the conclusion.

Changes to text: This qualitative study, nested within a cohort study assessing the outcomes of a treatment intervention, is the first to explore how women with GTPS manage daily life and how they experience the combined treatment of education, ESWT and exercise to understand the mechanisms possibly explaining their outcomes.

Pg 23, review wording ‘are important knowledge…’ to enhance clarity

Authors: We have reworded for clarity.

Changes to text: These new findings regarding how women with GTPS experienced and managed pain in daily life, and how they experience and adapt to treatment elements, are important improvements to inform clinical practice for women with GTPS.

6. PLOS authors have the option to publish the peer review history of their article (what does this mean?). If published, this will include your full peer review and any attached files.

Do you want your identity to be public for this peer review? For information about this choice, including consent withdrawal, please see our Privacy Policy.

Reviewer #1: Yes: Helen French

---

## [Decision Letter · Decision Letter 1]

2 May 2022

PONE-D-21-27762R1"I feel I have been taken seriously" Women’s experience of greater trochanteric pain syndrome treatment - a nested qualitative studyPLOS ONE

Dear Dr..Andreasen,

Thank you for submitting your manuscript to PLOS ONE. After careful consideration, we feel that it has merit but does not fully meet PLOS ONE’s publication criteria as it currently stands. Therefore, we invite you to submit a revised version of the manuscript that addresses the points raised during the review process.

Please make the suggested minor revisions.  In particular please take note of the cautions by reviewer one regarding generalizing from your results in your recommendations.

We look forward to receiving your revised manuscript.

Kind regards,

Rosemary Frey

Academic Editor

PLOS ONE

Journal Requirements:

Reviewers' comments:

Reviewer's Responses to Questions

**Comments to the Author**

1. If the authors have adequately addressed your comments raised in a previous round of review and you feel that this manuscript is now acceptable for publication, you may indicate that here to bypass the “Comments to the Author” section, enter your conflict of interest statement in the “Confidential to Editor” section, and submit your "Accept" recommendation.

Reviewer #1: All comments have been addressed

Reviewer #2: (No Response)

2. Is the manuscript technically sound, and do the data support the conclusions?

Reviewer #1: Yes

Reviewer #2: Partly

3. Has the statistical analysis been performed appropriately and rigorously? 

Reviewer #1: Yes

Reviewer #2: Yes

4. Have the authors made all data underlying the findings in their manuscript fully available?

Reviewer #1: No

Reviewer #2: No

5. Is the manuscript presented in an intelligible fashion and written in standard English?

Reviewer #1: Yes

Reviewer #2: Yes

6. Review Comments to the Author

Reviewer #1: Dear authors, many thanks for submitting this revision and addressing previous feedback. I have some minor suggestions only, some of which are more related to wording/language and grammar, based on some of the additional text added to the revision.

Pg 5, lines 98-100, this sentence which has been added by the authors needs a review in relation to clarity

The thorough introduction consisted of an explanation to the findings and diagnose of the individual participant at inclusion by CM

Suggest (if I understand correctly that this is what you mean?)

'This introduction consisted of an explanation regarding the clinical findings and diagnosis of each participant at inclusion by CM'

The inserted paragraph could be more succinctly written as suggested here

During the first session by JEJ, which included the ESWT intervention, participants were once more informed, including the provision of written information, about the known mechanisms of how GTPS and associated pain may be managed in everyday life. Participants were educated on how to avoid movements and situations causing pain (10, 28). The education was repeated and individualised for each patient´s experience, during the following two treatment sessions. At the six-week follow up, participant experience was addressed, and any positive or negative pain experiences were used to facilitate further pain management strategies specific for the individual participant. Thereby moving from a “general” education towards a customized “personal” education specific to the individual participant´s daily context and way of life.

Table 1: final column heading should read ‘pain relief’ rather than ‘pain relieve’. Were those descriptors based on some form of global rating of change scale? If so, please add a legend to explain it and provide a supporting reference for the scale.

In column 4, job situation, suggest the word unemployed (if that’s what ‘not in job’ means), as used in the text (line 153). Align the legend (line 157) under the table.

Pg 16, I wonder if the term ‘reduce activity levels’ is better here than ‘slow down’ (just a suggestion)

Pg 20, lines 347-349. I think sentence either needs to be two sentences or separated by a semi-colon

'Pain education and management was a central element of the intervention for the women as a measure of success was their ability to minimize pain and manage their daily life, during and after treatment'.

Pg 21, lines 378-381. I don’t think changing the word here to ‘professional’ is appropriate here, unless participants specifically mentioned the professional behaviour of the physiotherapist (which I did not see in any illustrative quotes).

I suggest the following

‘The therapeutic alliance, characterized by acknowledgement of the individuals’ situation, addressing individuals’ needs, and a secure therapeutic relationship, may be the mechanisms that supported the development of intrinsic motivation and self-efficacy as described by Ryan and Deci (42)’.

Pg 22, line 385-387. Suggest rewording that sentence to the following

'Both physiotherapists in our intervention were experienced specialists in managing GTPS, and the effects may not be directly transferable to other settings where the treating physiotherapists are not specialists in the management of this condition.'

Pg 24, line 446, Suggest removing the word ‘Especially’ before ‘Thorne’

Pg 25, line 476, Suggest the wording ‘additions to the current evidence’ is preferred than ‘improvements’

Check all quotes in the text and Table 3 to ensure consistency in formatting e.g. quotation marks, ID number in brackets.

Reviewer #2: Abstract:

Background: change to: “A combination of education, ESWT and exercise are…”. Line 24-26 should not be in the Background section of the abstract. Rather, use this space to explain why the aim is to look at the experiences of ‘their daily life’ (in addition to treatment).

Methods: I would suggest including important qualitative aspects like: were the interviews in-person? Were they video/audio recorded.

Rather than including ‘a six-step approach’, I would suggest spelling out the 6 steps, or change to something like: “transcripts were coded and analysed during a inductive thematic approach.”

Findings: include numbers of themes. (1)xx, (2) xxx, (3) xxx.

Line 36-40 mostly repeats the themes identified in line 33-36. Further, the names of the themes should/could be self-explanatory. I would suggest rewriting this section.

Conclusion: Lin 41-43 suggest that the aim was to explore (1) the experience and management of hip pain, and (2) experience and adapting to treatment. Firstly, I still do not quite understand why authors are exploring: daily living & experiences of treatment. Do authors mean that the daily living is explored as a result of the treatment? Or in general? If it is in general, I think authors should make a clearer rational to why these questions are combined in 1 research question. Secondly, please ensure your background, objectives, results, conclusions align with the specific research aim. Line 43-45: The reviewer would suggest to remove this section, as this is not a conclusion that is closely aligned with your findings.

Introduction

Aim: Line 81-83- objective; do authors mean that they explore experiences, feelings or perceptions of the impact of hip pain due to GTPS on daily life? Authors should be clear in what it is they are assessing. Authors should be clearer in the aim that this study looked at women with GTPS that were included in a cohort study/ after receiving treatment as part of a larger study, and that this qual study mostly reflects on this cohort study.

Methods:

Line 120: could you include more information about the background of the qualitative researcher?

Line 121: was the topic guide developed a priori? What did it include (questions, prompts, both)?

Did recruitment, data collection and analysis proceed concurrently? How did authors ensure data saturation?

Results:

The results section is very long and wordy (page 10-16). The reviewer suggests that authors rewrite the results section by integrating quotes in the theme explanations, to minimise repetition of findings.

Conclusion:

Line 460-461: this study explored experiences, as such, it did not “understand the mechanisms possibly explaining their outcomes”.

Authors should be cautious in inferring that findings can be used as a framework for an individualized patient education, treatment and evaluation strategy. What is this based on?

The conclusion should make clear that findings are in the context of the cohort study, and the specific treatment that the participants received.

7. PLOS authors have the option to publish the peer review history of their article (what does this mean?). If published, this will include your full peer review and any attached files.

Reviewer #1: **Yes: **Helen P French

Reviewer #2: No

---

## [Author Response · Author response to Decision Letter 1]

14 Jun 2022

Response to reviewers and editor

Thank you very much for the constructive comments from editor, reviewer 1 and reviewer 2, and for the opportunity to revise and improve the manuscript. As recommended, we have responded to each point raised by the reviewers before uploading a 'Revised Manuscript with Track Changes' as well as a manuscript without track changes labeled 'Manuscript'. 

We have addressed each point raised by the reviewers separately with the following structure:

1. Reviewer comment

2. Author response and actions

Comments to the Author

1. If the authors have adequately addressed your comments raised in a previous round of review and you feel that this manuscript is now acceptable for publication, you may indicate that here to bypass the “Comments to the Author” section, enter your conflict of interest statement in the “Confidential to Editor” section, and submit your "Accept" recommendation.

Reviewer #1: All comments have been addressed

Reviewer #2: (No Response)

2. Is the manuscript technically sound, and do the data support the conclusions?

Reviewer #1: Yes

Reviewer #2: Partly

3. Has the statistical analysis been performed appropriately and rigorously? 

Reviewer #1: Yes

Reviewer #2: Yes

4. Have the authors made all data underlying the findings in their manuscript fully available?

Reviewer #1: No

Reviewer #2: No

5. Is the manuscript presented in an intelligible fashion and written in standard English?

Reviewer #1: Yes

Reviewer #2: Yes

6. Review Comments to the Author

Reviewer #1: Dear authors, many thanks for submitting this revision and addressing previous feedback. I have some minor suggestions only, some of which are more related to wording/language and grammar, based on some of the additional text added to the revision.

Author response and actions: Thank you very much for your response and your suggestions, we have addressed each of your comments below.

Pg 5, lines 98-100, this sentence which has been added by the authors needs a review in relation to clarity

The thorough introduction consisted of an explanation to the findings and diagnose of the individual participant at inclusion by CM

Suggest (if I understand correctly that this is what you mean?)

'This introduction consisted of an explanation regarding the clinical findings and diagnosis of each participant at inclusion by CM'

Author response and actions: Thank you, we have revised as suggested

The inserted paragraph could be more succinctly written as suggested here

During the first session by JEJ, which included the ESWT intervention, participants were once more informed, including the provision of written information, about the known mechanisms of how GTPS and associated pain may be managed in everyday life. Participants were educated on how to avoid movements and situations causing pain (10, 28). The education was repeated and individualised for each patient´s experience, during the following two treatment sessions. At the six-week follow up, participant experience was addressed, and any positive or negative pain experiences were used to facilitate further pain management strategies specific for the individual participant. Thereby moving from a “general” education towards a customized “personal” education specific to the individual participant´s daily context and way of life.

Author response and actions: We agree and have revised as suggested.

Table 1: final column heading should read ‘pain relief’ rather than ‘pain relieve’. Were those descriptors based on some form of global rating of change scale? If so, please add a legend to explain it and provide a supporting reference for the scale.

In column 4, job situation, suggest the word unemployed (if that’s what ‘not in job’ means), as used in the text (line 153). Align the legend (line 157) under the table.

Author response and actions: Thank you, we have changed to “pain relief” and “unemployed”. We did not use a specific global rating scale, we asked the women about it and put them into one of these four categories according to their answers.

We have therefore inserted and aligned the legend under the table with the following:

# The women were asked how they experienced their improvement, and their response was categorized into one of four categories; no improvement, minor to moderate improvement, considerable improvement or full or nearly full recovery.

Pg 16, I wonder if the term ‘reduce activity levels’ is better here than ‘slow down’ (just a suggestion)

Author response and actions: We agree and have revised

Pg 20, lines 347-349. I think sentence either needs to be two sentences or separated by a semi-colon

'Pain education and management was a central element of the intervention for the women as a measure of success was their ability to minimize pain and manage their daily life, during and after treatment'.

Author response and actions: We agree and have inserted a semi-colon.

Pg 21, lines 378-381. I don’t think changing the word here to ‘professional’ is appropriate here, unless participants specifically mentioned the professional behaviour of the physiotherapist (which I did not see in any illustrative quotes).

I suggest the following

‘The therapeutic alliance, characterized by acknowledgement of the individuals’ situation, addressing individuals’ needs, and a secure therapeutic relationship, may be the mechanisms that supported the development of intrinsic motivation and self-efficacy as described by Ryan and Deci (42)’.

Author response and actions: Thank you for your suggestion, we have changed accordingly.

Pg 22, line 385-387. Suggest rewording that sentence to the following

'Both physiotherapists in our intervention were experienced specialists in managing GTPS, and the effects may not be directly transferable to other settings where the treating physiotherapists are not specialists in the management of this condition.'

Author response and actions: Thank you, we have reworded as suggested.

Pg 24, line 446, Suggest removing the word ‘Especially’ before ‘Thorne’

Author response and actions: We agree and have removed ‘Especially’.

Pg 25, line 476, Suggest the wording ‘additions to the current evidence’ is preferred than ‘improvements’

Author response and actions: We fully agree and have reworded the sentence for clarity.

Check all quotes in the text and Table 3 to ensure consistency in formatting e.g. quotation marks, ID number in brackets.

Author response and actions: We have scrutinized to ensure consistency in Table 3 and have put ID numbers in brackets, thank you.

Reviewer #2: Abstract:

Background: change to: “A combination of education, ESWT and exercise are…”. Line 24-26 should not be in the Background section of the abstract. Rather, use this space to explain why the aim is to look at the experiences of ‘their daily life’ (in addition to treatment).

Author response and actions: We have discussed this in the author group, and we find this is an important information in advance. However, we also find it important to explain why the aim is to look at the experiences of their daily life as well and we have therefore added this (The abstract is still less than 300 words).

Methods: I would suggest including important qualitative aspects like: were the interviews in-person? Were they video/audio recorded.

Author response and actions: We have added these aspects.

Rather than including ‘a six-step approach’, I would suggest spelling out the 6 steps, or change to something like: “transcripts were coded and analysed during a inductive thematic approach.”

Author response and actions: We have reworded as suggested.

Findings: include numbers of themes. (1)xx, (2) xxx, (3) xxx.

Author response and actions: We have included numbers of themes as suggested.

Line 36-40 mostly repeats the themes identified in line 33-36. Further, the names of the themes should/could be self-explanatory. I would suggest rewriting this section.

Author response and actions: We agree and have removed most of the text after numbering and naming the themes.

Conclusion: Lin 41-43 suggest that the aim was to explore (1) the experience and management of hip pain, and (2) experience and adapting to treatment. Firstly, I still do not quite understand why authors are exploring: daily living & experiences of treatment. Do authors mean that the daily living is explored as a result of the treatment? Or in general? If it is in general, I think authors should make a clearer rational to why these questions are combined in 1 research question. Secondly, please ensure your background, objectives, results, conclusions align with the specific research aim. Line 43-45: The reviewer would suggest to remove this section, as this is not a conclusion that is closely aligned with your findings.

Author response and actions: We hope our introduction in the abstract does clarify our reasoning – we did take a holistic approach and wanted to include contextual experiences of daily life and how the women managed their condition as the starting point for understanding their experiences with the treatment program. We have chosen to keep the last sentence in the conclusion however we have reworded to align this more with our findings.

Introduction

Aim: Line 81-83- objective; do authors mean that they explore experiences, feelings or perceptions of the impact of hip pain due to GTPS on daily life? 

Author response and actions: We hope that our previous explanations clarify that we reasoned that we would achieve more insightful and contextualised knowledge from the women by the approach chosen.

Authors should be clear in what it is they are assessing. Authors should be clearer in the aim that this study looked at women with GTPS that were included in a cohort study/ after receiving treatment as part of a larger study, and that this qual study mostly reflects on this cohort study.

Author response and actions: 

We agree that we could clarify this further. In the introduction we write the following that we “need to better understand how they experience their daily life with GTPS, how participation in a multi-modal treatment program is experienced by the women, and how it may improve their situation regarding pain and functional level.” 

We have clarified this by adding it to the aim as well: The aim of this study was to explore how women with greater trochanteric pain syndrome, participating in a multi-modal treatment program, experience and manage both their daily life and the combined treatment of education, ESWT and exercise.

Methods:

Line 120: could you include more information about the background of the qualitative researcher?

Author response and actions: We have revised the Methods section and added more information. We have added information about the qualitative researcher to the manuscript.

Line 121: was the topic guide developed a priori? What did it include (questions, prompts, both)?

Author response and actions: The topic guide was developed a priori. The interview guide included questions and prompts were used if necessary, to support or guide the participants. This is added to the manuscript.

Did recruitment, data collection and analysis proceed concurrently? 

Author response and actions: Recruitment and data collection was finalized before the analysis was initiated. This is also added to the manuscript.

How did authors ensure data saturation?

Author response and actions: As described in the methods section we strived to ensure rich data descriptions but did not strive for data saturation. We further discuss this in the discussion section. We sincerely hope that our explanations and references clarify this approach, an approach that is accepted within thematic analysis as it is questioned whether it is possible to be sure and as a researcher guarantee that you have covered all aspects of a topic. We have made no further changes to the manuscript after we revised the methodological considerations in the discussion section after the first revision as recommended by reviewer 1.

Results:

The results section is very long and wordy (page 10-16). The reviewer suggests that authors rewrite the results section by integrating quotes in the theme explanations, to minimise repetition of findings.

Author response and actions: We have scrutinized the results section and revised and condensed to some degree, however we find that the validation of the findings is a very important part of the results section as we find that this gives a deeper insight in to the lives of the female participants in the study. Further reviewer 1 has not commented further on the results section after the first revision was submitted to PLOS ONE. We sincerely hope that this approach is accepted.

Conclusion:

Line 460-461: this study explored experiences, as such, it did not “understand the mechanisms possibly explaining their outcomes”.

Author response and actions: Thank you for this comment. We do explore the experiences and perspectives of the women, but we did also use this knowledge to try to understand mechanisms possibly explaining their outcomes. We do state “possibly” as we do not provide any evidence for causal explanations in this nested qualitative study. However, we have revised and removed the last part of the sentence.

Authors should be cautious in inferring that findings can be used as a framework for an individualized patient education, treatment and evaluation strategy. What is this based on?

Author response and actions: Thank you for this response. We agree that we should be a little more cautious than to suggest this knowledge as sufficient evidence to suggest it used as a framework. We have therefore revised the last sentence in the conclusion.

The conclusion should make clear that findings are in the context of the cohort study, and the specific treatment that the participants received.

Author response and actions: We fully agree and have cautiously done this throughout the manuscript. We have made it clear in the abstract, in the aim, in the methods- and discussion section and finally it is stated in the start of the conclusion.

7. PLOS authors have the option to publish the peer review history of their article (what does this mean?). If published, this will include your full peer review and any attached files.

Do you want your identity to be public for this peer review? For information about this choice, including consent withdrawal, please see our Privacy Policy.

Reviewer #1: Yes: Helen P French

Reviewer #2: No

---

## [Decision Letter · Decision Letter 2]

19 Sep 2022

PONE-D-21-27762R2"I feel I have been taken seriously" Women’s experience of greater trochanteric pain syndrome treatment - a nested qualitative studyPLOS ONE

Dear Dr.Andreasen, 

Thank you for submitting your manuscript to PLOS ONE. After careful consideration, we feel that it has merit but does not fully meet PLOS ONE’s publication criteria as it currently stands. Therefore, we invite you to submit a revised version of the manuscript that addresses the points raised during the review process.

We look forward to receiving your revised manuscript.

Kind regards,

Rosemary Frey

Academic Editor

PLOS ONE

Journal Requirements:

Additional Editor Comments:

Please make the minor corrections and additions as suggested by Reviewer 3.

Reviewers' comments:

Reviewer's Responses to Questions

**Comments to the Author**

1. If the authors have adequately addressed your comments raised in a previous round of review and you feel that this manuscript is now acceptable for publication, you may indicate that here to bypass the “Comments to the Author” section, enter your conflict of interest statement in the “Confidential to Editor” section, and submit your "Accept" recommendation.

Reviewer #1: All comments have been addressed

Reviewer #3: (No Response)

2. Is the manuscript technically sound, and do the data support the conclusions?

Reviewer #1: Yes

Reviewer #3: (No Response)

3. Has the statistical analysis been performed appropriately and rigorously? 

Reviewer #1: Yes

Reviewer #3: N/A

4. Have the authors made all data underlying the findings in their manuscript fully available?

Reviewer #1: Yes

Reviewer #3: No

5. Is the manuscript presented in an intelligible fashion and written in standard English?

Reviewer #1: Yes

Reviewer #3: Yes

6. Review Comments to the Author

Reviewer #1: I have no further comments- well done to the authors who have addressed previous comments sufficiently.

Reviewer #3: Comments to author

Thank you for this study which adds knowledge about the experiences of women with GTPS.

Abstract

Background: Line 27: in the sentence “Exploring women’s experience … should help understand … “ suggest changing the word should to may

Methods: Line 34: in the sentence “Transcripts were coded and analyse during an …” suggest changing the word during to using

Introduction

Line 50 spell out GTPS in full when at beginning of sentence

Line 60: change the wording of “therefore been disproven” to softer response along the lines of “this may not be the case” recent evidence seems to support this is not the case etc.

Lines 70-74: I can see why you are introducing therapeutic alliance but feel this section needs a little more justification to include here and why it is of interest in your study.

Methods

Line 88: is the protocol for the cohort study published? If so, could reference here.

Line 42: you have chosen social constructive approach. Perhaps more information needed her as to why this was the most appropriate choice rather than others used in qualitative health studies such as qualitative content approach or phenomenological approach.

Lines 170-173: The 5 themes listed. I suggest you consider re-naming themes 3 and 5 as the way they currently are is just a summary of the findings rather than alluding to the direction of the theme. Suggest re-naming 3 and 5 so they better reflect the findings contained within these themes. Eg. theme 3, information helps/reassures/motivates is the general feeling I am getting; theme 5, women want more is the general message I am reading?

Discussion

Line 408: How? Provide some suggestions for the reader for how exercise may be addressed more carefully or some relevant strategies/directions they could take away

Limitations

The sample has experienced symptoms of varying duration ( 6 months – 10 years) and varying pain (3-8 rating) – I would assume this also has impacts upon the findings and may deserve a mention.

Also, a comment in the discussion about whether your theoretical approach was useful or had any limitations may be beneficial?

7. PLOS authors have the option to publish the peer review history of their article (what does this mean?). If published, this will include your full peer review and any attached files.

Reviewer #1: **Yes: **Helen P French

Reviewer #3: No

---

## [Author Response · Author response to Decision Letter 2]

24 Sep 2022

September 2022

Response letter,

Thank you very much for the constructive comments from the reviewers and for the opportunity to revise and improve the manuscript. As recommended, we have responded to each point raised by the reviewers before uploading a 'Revised Manuscript with Track Changes' as well as a manuscript without track changes labeled 'Manuscript'. 

We have addressed each point raised by the reviewers separately with the following structure:

1. Reviewer comment

2. Author response and actions

6. Review Comments to the Author

Reviewer #1: I have no further comments- well done to the authors who have addressed previous comments sufficiently.

Response: Thank you very much.

Reviewer #3: Comments to author

Thank you for this study which adds knowledge about the experiences of women with GTPS.

Response: Thank you very much.

Abstract

Background: Line 27: in the sentence “Exploring women’s experience … should help understand … “ suggest changing the word should to may

Response: We have revised to the suggested change

Methods: Line 34: in the sentence “Transcripts were coded and analyse during an …” suggest changing the word during to using

Response: We have revised as suggested

Introduction

Line 50 spell out GTPS in full when at beginning of sentence

Response: Revised 

Line 60: change the wording of “therefore been disproven” to softer response along the lines of “this may not be the case” recent evidence seems to support this is not the case etc.

Response: We have revised as suggested to a softer response

Lines 70-74: I can see why you are introducing therapeutic alliance but feel this section needs a little more justification to include here and why it is of interest in your study.

Response: We have added further justification for the focus on therapeutic alliance.

Changes to text: This is important to successfully addressing GTPS, given that all successful interventions have included elements of education and exercise that require active participation, learning and behavior change. Therefore, aspects of therapeutic alliance as a contributor to the experiences of effect and confidence in managing GTPS are important to explore further.

Methods

Line 88: is the protocol for the cohort study published? If so, could reference here.

Response: We agree, but the protocol has not been published. The findings of the cohort study are being processed for publication at the moment.

Line 42: you have chosen social constructive approach. Perhaps more information needed her as to why this was the most appropriate choice rather than others used in qualitative health studies such as qualitative content approach or phenomenological approach.

Response: We have added information on why we find this approach beneficial.

Changes to text:

The theoretical approach was social constructivism to explore insights regarding how people interact with their social world, and focused on the participants’ experiences in their particular contexts (33,35). This focus can therefore help us to explore how the women themselves construct their knowledge, understanding and management of the GTPS condition in their everyday life.

Lines 170-173: The 5 themes listed. I suggest you consider re-naming themes 3 and 5 as the way they currently are is just a summary of the findings rather than alluding to the direction of the theme. Suggest re-naming 3 and 5 so they better reflect the findings contained within these themes. Eg. theme 3, information helps/reassures/motivates is the general feeling I am getting; theme 5, women want more is the general message I am reading?

Response: We have re-named theme 3 and 5 as suggested by the reviewer as we have pointed more to the direction of the theme.

Theme 3: The participants´ experiences of the intervention – information is key

Theme 5: The women´s perspectives on improving and expanding the intervention

Discussion

Line 408: How? Provide some suggestions for the reader for how exercise may be addressed more carefully or some relevant strategies/directions they could take away

Response: We have added suggestions

Changes to text:

The issues related to non-adherence and low compliance is well-known in general (43) and in physiotherapy as well regarding non-supervised exercises (44,45) and should be addressed carefully, i.e by a thorough dialogue with the women regarding the type and intensity of pain being acceptable during exercise as well as carefully selecting which and how many exercises to provide to each woman.

Limitations

The sample has experienced symptoms of varying duration ( 6 months – 10 years) and varying pain (3-8 rating) – I would assume this also has impacts upon the findings and may deserve a mention.

Response: Thank you for this comment, we agree, this deserves a mention. We have added this.

Changes to text:

Finally, it should be noticed that the women have experienced symptoms of varying duration and varying pain.

Also, a comment in the discussion about whether your theoretical approach was useful or had any limitations may be beneficial?

Response: We agree and have added this to the section:

Changes to text:

We find the theoretical approach in this study useful as the analysis of the data using this approach helped us to explore and emphasize that the women individually constructed their understanding and management of the GTPS condition in their given contexts. 

On behalf of the author group,

Jane Andreasen

---

## [Decision Letter · Decision Letter 3]

31 Oct 2022

PONE-D-21-27762R3"I feel I have been taken seriously" Women’s experience of greater trochanteric pain syndrome treatment - a nested qualitative studyPLOS ONE

Dear Dr. Andreasen,

Thank you for submitting your manuscript to PLOS ONE. After careful consideration, we feel that it has merit but does not fully meet PLOS ONE’s publication criteria as it currently stands. Therefore, we invite you to submit a revised version of the manuscript that addresses the points raised during the review process.

Please submit your revision by Dec 15 2022 11:59PM. Contact the journal office at plosone@plos.org. Please include the following items when submitting your revised manuscript:A rebuttal letter that responds to each point raised by the academic editor and reviewer(s). You should upload this letter as a separate file labeled 'Response to Reviewers'.A marked-up copy of your manuscript that highlights changes made to the original version. You should upload this as a separate file labeled 'Revised Manuscript with Track Changes'.An unmarked version of your revised paper without tracked changes. You should upload this as a separate file labeled 'Manuscript'.If applicable, we recommend that you deposit your laboratory protocols in protocols.io to enhance the reproducibility of your results. Protocols.io assigns your protocol its own identifier (DOI) so that it can be cited independently in the future. For instructions see: https://journals.plos.org/plosone/s/submission-guidelines#loc-laboratory-protocols. Additionally, PLOS ONE offers an option for publishing peer-reviewed Lab Protocol articles, which describe protocols hosted on protocols.io. Read more information on sharing protocols at https://plos.org/protocols?utm_medium=editorial-email&utm_source=authorletters&utm_campaign=protocols.

We look forward to receiving your revised manuscript.

Kind regards,

Rosemary Frey

Academic Editor

PLOS ONE

Journal Requirements:

Additional Editor Comments:

Please update the theme titles as per the request of reviewer 3.

Reviewers' comments:

Reviewer's Responses to Questions

**Comments to the Author**

1. If the authors have adequately addressed your comments raised in a previous round of review and you feel that this manuscript is now acceptable for publication, you may indicate that here to bypass the “Comments to the Author” section, enter your conflict of interest statement in the “Confidential to Editor” section, and submit your "Accept" recommendation.

Reviewer #1: (No Response)

Reviewer #3: (No Response)

2. Is the manuscript technically sound, and do the data support the conclusions?

Reviewer #1: (No Response)

Reviewer #3: Yes

3. Has the statistical analysis been performed appropriately and rigorously? 

Reviewer #1: (No Response)

Reviewer #3: Yes

4. Have the authors made all data underlying the findings in their manuscript fully available?

Reviewer #1: (No Response)

Reviewer #3: No

5. Is the manuscript presented in an intelligible fashion and written in standard English?

Reviewer #1: (No Response)

Reviewer #3: Yes

6. Review Comments to the Author

Reviewer #1: (No Response)

Reviewer #3: Thank you for addressing comments.

Please update the new theme titles for consistency throughout the paper - in abstract results section and in paper lines 175-178 where the themes are listed.

7. PLOS authors have the option to publish the peer review history of their article (what does this mean?). If published, this will include your full peer review and any attached files.

Reviewer #1: **Yes: **Helen French

Reviewer #3: No

---

## [Author Response · Author response to Decision Letter 3]

1 Nov 2022

Dear editor Rosemary Frey and reviewers,

Thank you very much for letting me revise again. I want to apologize for the not fully revised manuscript that I forwarded to you.

6. Review Comments to the Author

Reviewer #1: (No Response)

Reviewer #3: Thank you for addressing comments.

Please update the new theme titles for consistency throughout the paper - in abstract results section and in paper lines 175-178 where the themes are listed.

Response: I apologize once again for the inconvenience and I have revised as suggested, both in the abstract and in the manuscript.

Kind regards,

Jane Andreasen

---

## [Decision Letter · Decision Letter 4]

14 Nov 2022

"I feel I have been taken seriously" Women’s experience of greater trochanteric pain syndrome treatment - a nested qualitative study

PONE-D-21-27762R4

Dear Dr Andreasen, 

We’re pleased to inform you that your manuscript has been judged scientifically suitable for publication and will be formally accepted for publication once it meets all outstanding technical requirements.

Kind regards,

Rosemary Frey

Academic Editor

PLOS ONE

Additional Editor Comments (optional):

Reviewers' comments:

Reviewer's Responses to Questions

**Comments to the Author**

1. If the authors have adequately addressed your comments raised in a previous round of review and you feel that this manuscript is now acceptable for publication, you may indicate that here to bypass the “Comments to the Author” section, enter your conflict of interest statement in the “Confidential to Editor” section, and submit your "Accept" recommendation.

Reviewer #3: All comments have been addressed

2. Is the manuscript technically sound, and do the data support the conclusions?

Reviewer #3: Yes

3. Has the statistical analysis been performed appropriately and rigorously? 

Reviewer #3: Yes

4. Have the authors made all data underlying the findings in their manuscript fully available?

Reviewer #3: No

5. Is the manuscript presented in an intelligible fashion and written in standard English?

Reviewer #3: Yes

6. Review Comments to the Author

Reviewer #3: (No Response)

7. PLOS authors have the option to publish the peer review history of their article (what does this mean?). If published, this will include your full peer review and any attached files.

Reviewer #3: No

---

## [Editor Report · Acceptance letter]

17 Nov 2022

PONE-D-21-27762R4 

*“I feel I have been taken seriously”*
**Women’s experience of greater trochanteric pain syndrome treatment - a nested qualitative study**

Dear Dr. Andreasen:

I'm pleased to inform you that your manuscript has been deemed suitable for publication in PLOS ONE. Congratulations! Your manuscript is now with our production department. 

Kind regards, 

on behalf of

Dr. Rosemary Frey 

Academic Editor

PLOS ONE